

# Machine learning approaches for automatic classification of single-particle mass spectrometry data

Guanzhong Wang[1], Heinrich Ruser[1], Julian Schade[2, 3], Johannes Passig[3, 4, 5], Thomas Adam[2, 4], Günther Dollinger[1], and Ralf Zimmermann[3, 4, 5]

[1]Institute for Applied Physics and Measurement Technology, University of the Bundeswehr Munich, Neubiberg, 85577, Germany
[2]Institute of Chemistry and Environmental Engineering, University of the Bundeswehr Munich, Neubiberg, 85577, Germany
[3]Joint Mass Spectrometry Centre, Chair of Analytical Chemistry, University of Rostock, Rostock, 18059, Germany
[4]Joint Mass Spectrometry Centre, Helmholtz Zentrum München, Neuherberg, 85764, Germany
[5]Department Life, Light & Matter, Interdisciplinary Faculty, University of Rostock, Rostock, 18059, Germany

Correspondence to: Heinrich Ruser (heinrich.ruser@unibw.de)

**Abstract.** The chemical composition of aerosol particles is a key parameter for human health and climate effects. Single-
particle mass spectrometry (SPMS) has evolved to a mature technology with unique chemical coverage and the capability to analyze the distribution of aerosol components in the particle ensemble in real-time. With the fully automated characterization of the chemical profile of the aerosol particles, selective real-time monitoring of air quality could be performed e.g. for urgent risk assessments due to particularly harmful pollutants. For aerosol particle classification, mostly unsupervised clustering algorithms (ART-2a, K-means and their derivatives) are used, which require manual post-
processing. In this work, we focus on supervised algorithms to tackle the problem of automatic classification of large amounts of aerosol particle data. Supervised learning requires data with labels to train a predictive model. Therefore, we created a labeled benchmark dataset containing ~24,000 particles with eight different coarse categories that were highly abundant at a measurement in summer in Central Europe: *Elemental Carbon (EC), Organic Carbon and Elemental Carbon (OC-EC), Potassium-rich (K-rich), Calcium-rich (Ca-rich), Iron-rich (Fe-rich), Vanadium-rich (V-rich), Magnesium-rich*
*(Mg-rich)* and *Sodium-rich (Na-rich)*. Using the chemical features of particles the performance of the following classical supervised algorithms was tested: K-nearest neighbors, support vector machine, decision tree, random forest and multi-layer perceptron. This work shows that despite the entrenched position of unsupervised clustering algorithms in the field, the use of supervised algorithms has the potential to replace the manual step of clustering algorithms in many applications, where real-time data analysis is essential. For the classification of the eight classes, the prediction accuracy of several supervised
algorithms exceeded 97 %. The trained model was used to classify ~49,000 particles from a blind dataset in 0.2 seconds, taking into account also a class of "unclassified" particles. The predictions are highly consistent with the results obtained in a previous study using ART-2a.



## 1 Introduction

In recent years, chemical aerosol particle analysis has received great attention from scientific communities and authorities, for its relevance for climate change, environmental pollution, and human health. However, particulate matter (PM) pollution control and management remain a huge challenge due to the complex physicochemical properties, sources and evolution of aerosol particles. An important indicator of air quality is the concentration of suspended particles in the air (usually particle mass via PM10 or PM2.5). Particles from different sources and with different chemical compositions are expected to cause various negative health effects (Dall'Osto & Harrison, 2006; Harrison & Yin, 2000; Maynard, 2004). A prolific method to obtain the size and chemical signatures of individual aerosol particles in real-time is single-particle mass spectrometry (SPMS) (Passig & Zimmermann, 2021; Pratt & Prather, 2012; Schade et al., 2019). From the particle's flight time between two laser beams, its size and velocity are derived and the proper time to trigger a laser shot for laser desorption/ionization (LDI) of the respective particle is calculated. After LDI decomposition, both positive and negative ions are separated by their mass. The resulting mass spectra are plots of the signal intensity versus the mass-to-charge ratio (m/z) of the ions (Anderson et al., 2005; Murphy, 2007) and can be understood as high-dimensional vectors. Since the aerosol particles from different sources can carry unique chemical characteristics and often retain these characteristics also after long-range transport (Dall'Osto & Harrison, 2006), the identification and classification of SPMS data can help to improve the understanding of regional PM and provide decision makers with the necessary information to determine effective control methods. Widely used classification methods in the SPMS community are unsupervised clustering algorithms, which require manual post-processing e.g. to select and re-merge resulting clusters. The primary target of this study is to develop alternative methods to perform automatic classification in order to achieve real-time monitoring of air quality, such as supervised learning methods which have been successfully used in various domains for complex data classification. Furthermore, the supervised data classification allows a rapid classification of the vast majority of "common" particles in ambient air, at a work place environment or in an air quality screening setup. Among the reduced residual particle ensemble, rare particles, which might be indicators for specific sources or for potential hazard, can be identified more easily by matching with library spectra of hazardous particles.

## 2 Related works

### 2.1 Mass spectra classification with unsupervised learning

Unsupervised learning clustering algorithms are commonly used for the classification of mass spectra. The classification is based on geometric relationships between the spectral vector and different cluster centers. Samples (i.e. mass spectra) belonging to a certain cluster would be more similar than samples classified to other clusters. K-means (MacQueen, 1967) cluster center is minimized. It requires the user to set in advance the number of clusters $K$ to be classified. Setting the optimal $K$ value is a major challenge, even though there are some techniques to help determine the relatively appropriate value, such




as the "elbow" method and the "gap" statistic (Tibshirani et al., 2001). The ART-2a algorithm (Carpenter et al., 1991) was

applied by (Song et al., 1999) to classify SPMS data. A parameter called "vigilance" is key to control the generated number of clusters. If the Cosine Similarity between a new sample (i.e. mass spectrum) and its nearest cluster center is greater than or equal to the preset "vigilance" value, it will be assigned to this cluster, with the position of its center shifted towards the new cluster member according to the "learning rate". If a new sample does not have enough similarity to all existing cluster centers, it will become a new cluster by itself. It is this dynamic network characteristic that allows ART-2a to discover new

categories in the data without disturbing existing ones.

In the field of SPMS data analysis, Arndt et al. (2021) and Healy et al. (2010, 2012) used twice K-means to classify the collected 558,740, 1.75 million and ~800,000 particles, respectively. For the first time particles are classified into dozens of clusters (50, 80 and 80), and then similar clusters are regrouped again to reduce the number of clusters (14, 15 and 33). Zelenyuk et al. (2006) added a distance threshold when using K-means to determine if a new sample should be classified

into any of the existing clusters, based on the same philosophy as the "vigilance" parameter used in the ART-2a algorithm. Dall'Osto & Harrison, (2006) used ART-2a to classify 128,290 particles into 490 clusters, then selected the particles from the top 200 clusters for analysis (the remaining particles were discarded) and reduced the total number of clusters by remerging similar clusters to five main clusters. In the same way, Dall'Osto et al. (2013); Dall'Osto & Harrison, (2012) and L. Li et al. (2014) classified the 153,595, 1,35 million and 510,341 particles with ART-2a and manually selected and re-

combined the generated clusters into 15, 10 and 5 clusters, respectively. Passig et al. (2022) applied a novel approach where individual particles are analyzed simultaneously by two different ionization techniques, i.e. by the classical LDI process (metals, salts, elements, EC, OC) as well as by laser desorption / resonance-enhanced multi photon ionization (LD-REMPI). This combination allows the single particle- detection of health relevant organic trace chemicals, in particular, carcinogenic polycyclic aromatic hydrocarbons (PAH). This complicates the situation, as particles may be either clustered according to

the LDI data or by the LD-REMPI MS PAH fingerprint. The authors focused on the PAH fingerprint and classified 4,412 PAH-containing particles into 733 clusters, and then merged the first 300 clusters into 10 PAH classes, which included ~85 % of all particles and could be associated with different sources. From the above studies, it can be noted that the number of clusters generated by clustering algorithms is usually much larger than their final number after selecting and re-merging. Generally, the more clusters are proposed, the higher the accuracy of the final results obtained after manual post-processing.

However, due to the large number of clusters the manual workload is high.

Other cluster algorithms applied with aerosol particle classification use hierarchical clustering that creates a hierarchical clustering tree by calculating the distance between mass spectra (Murphy et al., 2003; Rebotier & Prather, 2007). Still others use density-based clustering algorithms such as DBSCAN (Ester et al., 1996) and OPTICS (Ankerst et al., 1999) to classify the aerosol particles. The advantage of these algorithms is that they can divide regions with high enough density and find any

shapes of clusters and noise in the data. For example, if the data has a non-spherical distribution, the effect of using K-means will be greatly reduced. In this case, density-based algorithms would yield better results. Zhou et al. (2006) compared the performance of the classification of SPMS data with both ART-2a and DBSCAN, and Zhao et al. (2008) suggested to join



them. Reitz et al. (2016) used the results of OPTICS to set the number of classes needed for fuzzy c-means clustering to better process SPMS data.

The benefits of unsupervised learning are obvious: Their structures are relatively simple and the number of parameters that need to be tuned is small. Furthermore, through manual post-processing, the classification is a safe and conservative procedure where unknown particle classes and novel features are not easily overlooked. The discovery of new particle classes can help to update the database for the training of supervised learning models. However, unsupervised methods also have the following disadvantages: 1) They require manual post-processing. 2) Mass spectra within a cluster may not exactly
match each other chemically, even though they mathematically belong to that cluster (Murphy et al., 2003). 3) It's not easy to analyze the effect of different parameters on the results. Therefore, the same parameter configurations of ART-2a (vigilance, learning rate, number of iterations) were used and considered as "standard values" in some studies. In contrast, supervised learning shows improvements in those aspects.

## 2.2 Supervised learning

Supervised machine learning and deep learning algorithms have achieved tremendous success in many fields. In particular, neural network-based methods have revolutionized image processing by allowing machines to learn complex patterns and representations directly from the data. These techniques could also help to identify patterns in large SPMS datasets. Supervised learning requires a high-quality, balanced, and standardized dataset; imbalanced (biased) datasets will distort the performance (Bishop, 2006). Each sample in the dataset has a set of features and a classification label i.e. the classification
results are known a priori. Supervised algorithms learn the features to obtain a trained model, which can predict unlearned data automatically and does not require any manual post-processing. In this work, we tested the performance of several classical supervised algorithms for SPMS data classification. A brief description of the basic principles of the selected algorithms is given below. The performance of automatic classification of SPMS data will be described in Section 4.2 and Section 4.3.

**K-nearest neighbors (K-NN)** is a simple classification method without training. It determines the $K$ nearest neighbors of the new sample by calculating the distance between the new sample and each sample in the dataset, and classifies the new sample according to the most dominant category (label) among these $K$ nearest neighbors (Segaran, 2007). K-NN has similarity with clustering algorithms in that it is also distance-based.

**Support vector machine (SVM)** is a generalized linear classifier designed for binary classification problems. Its decision
boundary is the maximum-margin hyperplane of the samples, which means that the distance to the hyperplane of the data point closest to the hyperplane should be maximum (Awad & Khanna, 2015). The decision boundary can be extended from linear to nonlinear using different kernel functions (Noble, 2006). Multiple classification tasks are achieved by building multiple decision boundaries in an orderly manner using standard binary SVMs, usually based on one of two strategies of constructing classifiers: One-vs-All or One-vs-One (H. Li et al., 2005).



**Decision tree (DT)** is a non-parametric classification method with well-traceable decisions (Mitchell, 1997). DT starts from the root node and assigns each sample to one of its children nodes (leaves) and their leaves according to trained threshold values of certain parameters (features) forming an if-then tree structure of hierarchical parent-child relationships (Ge & Wong, 2008).

**Random forest (RF)** (Breiman, 2001) is a classifier consisting of multiple DTs ("forest") that are not associated with each 135 other. A new sample is judged separately by each DT in the forest. Compared to a single DT model, a RF consisting of a large number of unrelated DTs will produce more reliable predictions and be less prone to overfitting. Christopoulos et al. (2018) used RF for the classification of SPMS data of soil probes. The model was trained with 110 independent DTs, and for the classification of four classes (secondary organic aerosol, mineral dust, fertile soil and biological aerosols), a classification model with an accuracy of over 90 % was obtained.

**Multi-layer perceptron (MLP)** is a fully connected neural network consisting of an input layer, one or more hidden layers, and an output layer. Each neuron (node) carries an activation function, and the nodes in adjacent layers are connected by weighted edges (weights). Using fewer hidden layers makes the model less capable of learning features; more hidden layers, however, do not always lead to improved performance and usually increase the computational load (Ramchoun et al., 2016). The learning process consists of forward and backward propagation. Forward propagation is the process of computing the 145 output of each node using the activation function and weights. The loss is then calculated by the difference between the output value obtained from forward propagation and the actual value from the label. Backward propagation trains the neural network by computing the partial deviations (gradients) of each node in the opposite direction based on the loss function. These gradients are then fed to an optimization method to update the weights in the network in order to minimize the loss function (Ettaouil & Ghanou, 2009). In simple terms, backward propagation is like guiding the model to fix the mistakes it 150 made during forward propagation.

With all the presented supervised algorithms, fast fully automated classification of large amounts of SPMS data is feasible. However, whether to apply supervised or unsupervised learning will depend on the application scenario, bearing in mind the following challenges of supervised learning algorithms: 1) The process of creating a labeled dataset can be very time-consuming and expensive; publicly available labeled datasets are lacking. 2) Disambiguation: Some mass numbers have 155 multiple meanings in different situations, and often these mass peaks play a key role in classification, which makes pattern recognition more difficult, for example, $m/z = 24$ represents $Mg^+$ or $C_2^+$; $m/z = 39$ represents $K^+$, $NaO^+$ or $C_3H_3^+$; $m/z = 51$ represents $V^+$ or $C_4H_3^+$; $m/z = 56$ represents $Fe^+$, $CaO^+$ or $C_3H_4O^+$, etc. 3) Classes that are not present in the training data cannot be identified.

In this study, we demonstrate the capabilities of supervised algorithms to automatically classify aerosol particles. We created 160 a benchmark dataset with ~24,000 mass spectra. This dataset might serve as a resource for the development of new, efficient and accurate classification methods. We implemented and tested the performance of different algorithms for aerosol particle classification using their chemical compositions, i.e. positive and negative spectra. Our results provide researchers with an overview of the applicability of supervised machine learning algorithms to the classification of SPMS data and also provide



a basis for selecting appropriate algorithms. Prediction results from blind data show that the proposed use of supervised

learning is particularly well-suited for real-time specific particle detection.

## 3 Methodologies

### 3.1 Sampling site and equipment

To investigate the composition and possible sources of aerosol particles, especially emissions from ships, in the urban area of the coastal city of Rostock, Germany, a single-particle mass spectrometer was deployed from 26 June to 02 July 2018. The

sampling site was on the roof of a laboratory building at the University of Rostock, which is located in the southern part of the city (54°04'41.5''N, 12°06'30.6''E, about 35 meters above sea level). About 10 km to the north of the sampling site is the harbor of Rostock, and about 40 km to the north of the sampling site is the main shipping route. The town is remote from other large towns and industries, located at the sea in an agricultural region with forests. An overview of the campaign, the local meteorology and the instrumentation deployed is described in detail elsewhere (Passig et al., 2021). The SPMS

instrument, made by Photonion GmbH (Schwerin, Germany), consists of a dual-polarity TOF mass spectrometer in Z geometry. Continuous-wave lasers are used to detect the particles prior to LDI with a 248 nm KrF excimer laser (L. Li et al., 2011; Passig et al., 2020; Schade et al., 2019; Y. Zhou et al., 2016). During the campaign, 162.288 of the 290.144 particles detected by the SPMS instrument featured at least four significant peaks in their mass spectrum and were analyzed with respect to their chemical composition. From the mass spectra, 240 possible mass peaks (120 for each of the negative and

positive ions) were considered and each peak corresponds to a different mass-to-charge ratio (m/z).

### 3.2 Dataset

For image data, even untrained personnel can perform data annotation work (i.e. labeling the correct class affiliation) quickly and accurately. For SPMS data, considerable expertise is required from the annotators, which increases the difficulty of creating the dataset. To the best of our knowledge, there is no publicly available dataset of labeled atmospheric aerosol

particles. To test the performance of supervised algorithms for SPMS data classification, we built our own labeled dataset. In the process of manual labeling, we determined and labeled the particles based often on the highest ion peaks in the mass spectra, and applied the nomenclature of particle classes used corresponding to other sources (Ault et al., 2009; Dall'Osto & Harrison, 2006; Spencer et al., 2006). We divided the data from this campaign into two parts, one for labeling, one for verification ("blind data"). The data from four days (June 26-29) containing a total of 110,390 particles were used for

labeling. We selected and manually labeled 24,030 individual aerosol particle mass spectra from this part of data. Detailed information on the eight classes and their subclasses as well as the labeling rules and the number of samples are listed in Table 1 and Table 2. In Fig. 1, typical mass spectra of the eight aerosol particle classes are displayed. In the second part, we used all data from two consecutive days (June 30 to July 1) with 49,097 particles in total as "blind data", unrelated to the first part. In summary, the first part of the data is labeled data used to evaluate supervised learning algorithms, and



subsequently, those trained models were used to automatically classify mass spectra from the second part of the data
obtaining the temporal distribution of particle classes over a two-day period. The following is a description of the eight
coarse particle classes used in this work.

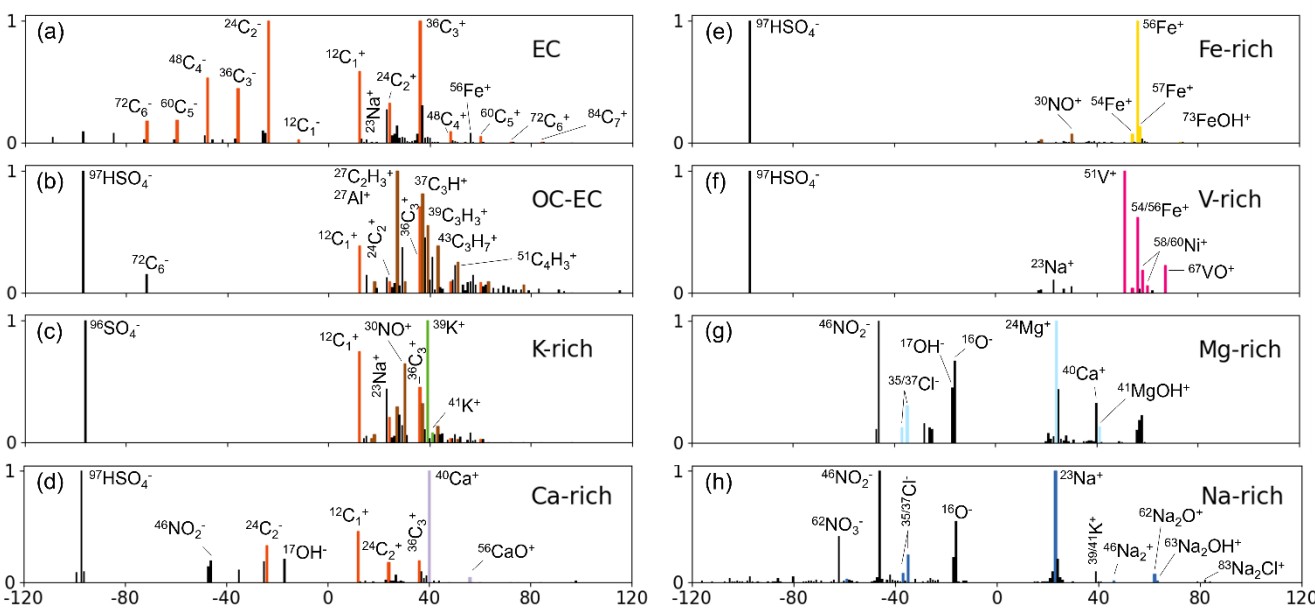

**Figure 1: Representative mass spectra of aerosol particles, attributed to one of eight classes of our labeled dataset. The positive and negative mass spectra are normalized separately according to their highest ion peaks. The highlighted ion signals are the signature peaks to distinguish different particles.**

*Elemental Carbon (EC)* signatures in particle mass spectra often result from any combustion source, but engines emit
particularly high numbers. This type of particles is predominantly observed in most SPMS studies (Ault et al., 2009;
Dall'Osto & Harrison, 2006; Healy et al., 2010; L. Li et al., 2014; Toner et al., 2006). A mass spectrum will be labeled as
belonging to the EC class when the entire mass spectrum is dominated by the EC signal and all other ion signals are weak.

*Organic carbon and elemental carbon (OC-EC)* particle sources are often associated with biomass burning due to
incomplete combustion. Other important emission sources are vehicles, ships, coal combustion, cooking, etc. (Furutani et al.,
2011; Healy et al., 2010; L. Li et al., 2014; Moffet et al., 2008; Shen et al., 2019). Xiao et al. (2018) have demonstrated that
fuel combustion emissions produce more EC and OC-EC than pure OC particles. In addition to incomplete combustion, OC
tends to adhere to EC, forming further OC-EC particles. Therefore, during labeling, pure OC particles were treated as a
subclass of the OC-EC class.




**Table 1: Overview of the eight classes in the created dataset. The ion markers used to label the mass spectra are summarized from various SPMS lab and field studies. The ions in the positive ions column are not the only signals contained in the positive mass spectrum but significant to differentiate them. The ions in the negative ions column are the ions that may be contained in that class.**

| Classes | Sub-classes | Possible sources | Positive ions | Negative ions | References |
|---|---|---|---|---|---|
| EC | - | Traffic, Biomass burning | $EC: 12n\,C_n^+$ | $EC: 12n\,C_n^-$; sulfate: $^{80}[SO_3]^-$, $^{96}[SO_4]^-$, $^{97}[HSO_4]^-$; sometimes no neg. signal | 1, 2, 3, 4, 5 |
| OC-EC | OC, OC-EC, OC-amine | Biomass burning, Traffic, Shipping | $OC: {}^{27}[C_2H_3]^+, {}^{37}[C_3H]^+, {}^{39}[C_3H_3]^+, {}^{43}[C_3H_7]^+, {}^{51}[C_4H_3]^+, {}^{63}[C_5H_3]^+, {}^{77}[C_6H_5]^+; EC;$ amine: ${}^{17}[NH_3]^+, {}^{18}[NH_4]^+, {}^{30}[NO]^+$ | $EC;$ sulfate; nitrate: ${}^{46}[NO_2]^-, {}^{62}[NO_3]^-$ | 1, 2, 5, 6, 7 |
| K-rich | K-EC, K-OC-EC, K-dominant, K-Cl, K-CN | Biomass burning | ${}^{39/41}K^+; EC; OC;$ amine | nitrate; sulfate; ${}^{35/37}Cl^-$; organonitrogen (CN): ${}^{26}[CN]^-$, ${}^{42}[CNO]^-$ | 1, 4, 5, 8, 9, 10, 11, 12 |
| Ca-rich | Ca-EC, Ca-Nit, Ca-Na-Fe | Lubrication oil from traffic or shipping, Dust | ${}^{40}Ca^+, {}^{56}[CaO]^+, {}^{57}[CaOH]^+, {}^{75}[CaCl]^+, {}^{96}[Ca_2O]^+$ | nitrate; sulfate; CN; ${}^{35/37}Cl^-$; EC; phosphate: ${}^{63}[PO_2]^-, {}^{79}[PO_3]^-, {}^{95}[PO_4]^-$ | 1, 2, 5, 6, 7, 8, 9, 13, 14 |
| Fe-rich | Fe-Nit, Fe-EC, Fe-dominant | Traffic, Shipping, Industry | ${}^{54/56/57}Fe^+, {}^{73}[FeOH]^+$ | nitrate; sulfate; EC; ${}^{16}O^-; {}^{17}[OH]^-;$ CN; ${}^{35/37}Cl^-$; | 1, 5, 15, 16, 17, 18, 19, 20 |
| V-rich | freshly and aged emitted | Shipping | ${}^{51}V^+, {}^{67}[VO]^+; {}^{54/56}Fe^+; {}^{58/60}Ni^+$ | sulfate; nitrate; EC; sometimes no neg. signal | 2, 3, 4, 14, 17, 20, 21 |
| Mg-rich | - | Sea Salt | ${}^{24}Mg^+, {}^{41}[MgOH]^+; {}^{23}Na^+; {}^{40}Ca^+, {}^{57}[CaOH]^+$ | ${}^{35/37}Cl^-$; nitrate; sulfate; ${}^{16}O^-; {}^{17}[OH]^-;$ CN | 1, 22 |
| Na-rich | - | Sea Salt | ${}^{23}Na^+, {}^{39}[NaO]^+, {}^{46}[Na_2]^+, {}^{62}[Na_2O]^+, {}^{63}[Na_2OH]^+, {}^{81/83}[Na_2Cl]^+$ | ${}^{35/37}Cl^-; {}^{59/61}[NaCl]^-, {}^{93/95}[NaCl_2]^-;$ nitrate; sulfate; ${}^{16}O^-; {}^{17}[OH]^-;$ CN | 1, 4, 5, 7, 12, 14 |

The reference numbers in the table are from the following publications: [1] (Dall'Osto & Harrison, 2006), [2] (Toner et al., 2006), [3] (Ault et al., 2009), [4] (Healy et al., 2010), [5] (L. Li et al., 2014), [6] (Dall'Osto & Harrison, 2012), [7] (Shen et al., 2019), [8] (Healy et al., 2012), [9] (Moffet et al., 2008), [10] (X. Liu et al., 2000), [11] (J. Li et al., 2003), [12] (Köllner et al., 2017), [13] (Shields et al., 2007), [14] (Passig et al., 2021), [15] (Arndt et al., 2021), [16] (Dall'Osto et al., 2016), [17] (Furutani et al., 2011), [18] (Passig et al., 2020), [19] (Gross et al., 2000), [20] (Passig et al., 2022), [21] (Z. Liu et al., 2017), [22] (L. Zhou et al., 2006)


**Table 2: Overview of the dataset. First column lists the total number (#) and percentage (%) of samples in the entire dataset and in each of the divided parts: training, validation and test set. The following columns represent the number and percentage of samples for the different classes.**

| Dataset | #; % | EC | OC-EC | K-rich | Ca-rich | Fe-rich | V-rich | Mg-rich | Na-rich |
|---|---|---|---|---|---|---|---|---|---|
| Total | 24030; 100 % | 4671; 19 % | 4000; 17 % | 3998; 17 % | 1365; 6 % | 1729; 7 % | 3879; 16 % | 540; 2 % | 3848; 16 % |
| Training | 14418; 60 % | 2803; 19 % | 2375; 16 % | 2404; 17 % | 816; 6 % | 1080; 7 % | 2321; 16 % | 349; 2 % | 2270; 16 % |
| Validation | 4806; 20 % | 916; 19 % | 812; 17 % | 766; 16 % | 269; 6 % | 338; 7 % | 777; 16 % | 100; 2 % | 828; 17 % |
| Test | 4806; 20 % | 952; 20 % | 813; 17 % | 828; 17 % | 280; 6 % | 311; 6 % | 781; 16 % | 91; 2 % | 750; 16 % |

***Potassium-rich (K-rich)*** particles have been identified in many studies as suitable tracers for both anthropogenic and natural biomass burning. In this study, the following subcategories are all assigned to the *K-rich* class: *K-EC-OC* particles were attributed to peat combustion (Healy et al., 2010) and are characterized by positive ion mass spectra containing high signals of potassium and sodium, as well as carbon and hydrocarbon fragment ions. *K-EC* particles might mainly come from local



traffic emissions and are often attributed to fossil fuel combustion (Dall'Osto & Harrison, 2006; L. Li et al., 2014). Wood
combustion particles exhibit a dominant signal of K$^+$, and we named this class *K-dominant* (Healy et al., 2010). *K-Cl*
particles were reported as a biomass combustion product and can also be attributed to cigarette smoke (Dall'Osto &
Harrison, 2006). Potassium and chlorine are initially combined or present in the liquid of vegetation (J. Li et al., 2003; X.
Liu et al., 2000). *K-CN* particles are also typical for biomass combustion (Dall'Osto & Harrison, 2012b; Köllner et al.,
2017), where CN represents the organo-nitrogen. The peak at *m/z = 39* may be the organic fragment $C_3H_3^+$ when it is not
much more intense than the other major hydrocarbon ion peaks. Potassium usually shows a more intense peak in the mass
spectrum than carbon cluster ions (Dall'Osto & Harrison, 2012; L. Li et al., 2014), so this is one of the bases for
distinguishing between *OC-EC* and *K-rich* classes when labeling.

***Calcium-rich (Ca-rich).*** Various studies (Dall'Osto et al., 2013; Dall'Osto & Harrison, 2012; Moffet et al., 2008; Passig et
al., 2021; Shields et al., 2007; Toner et al., 2006) have demonstrated that most soot particles from engines show calcium
characteristics from lubricant additives, potentially coming from emissions from vehicle traffic or ships. In addition, calcium
signal with silicon, silicon oxide or titanium dioxide are also evident in soil dust particles (Dall'Osto et al., 2016; Dall'Osto
& Harrison, 2006; L. Li et al., 2014; Shen et al., 2019).

***Iron-rich (Fe-rich)*** signatures are often combined with EC from anthropogenic combustion sources but may also be
associated with wear and tear of brake pads from traffic (Gross et al., 2000), may arise from biomass combustion (Chang-
Graham et al., 2011; Furutani et al., 2011) or industrial emissions (Arndt et al., 2021; L. Li et al., 2014). Note that the
resonant ionization of Fe at 248 nm laser wavelength increases the Fe signals compared to studies using other laser
wavelength (Passig et al., 2020).

***Vanadium-rich (V-rich)*** particles have a distinctive mass spectrum with peaks at V$^+$ and VO$^+$, and the combination of peaks
associated with the transition metals vanadium, iron and nickel is a well-documented signature of residual fuel combustion
particles from ship emissions (Ault et al., 2009; Furutani et al., 2011; Healy et al., 2010; Xiao et al., 2018). Some studies
(Ault et al., 2009; Z. Liu et al., 2017; Passig et al., 2021) discuss the differences in mass spectra of particles emitted from
ships with different degrees of aging, such as the predominance of sulfate as the negative signal in freshly emitted particles.
As the distance to the source increases, aged particles exhibit stronger nitrate or no negative spectral signal (Passig et al.,
2021). V-rich particles sometimes also show Ca$^+$ ions attributable to lubricant additives, as well as small signals from EC
and OC (Passig et al., 2021; Toner et al., 2006). If the positive mass spectra contain a V-Fe-Ni combination and their signals
are not the highest ion peaks, based on the difference in amplitude between these peaks and the highest ion peak we
determined, whether these mass spectra should be labeled as belonging to the V-rich class. For example, we labeled some
mass spectra as *V-rich* class, when they contain EC, OC, Ca$^+$ or Na$^+$ as the highest ion peaks and also have a certain intensity
of the combined V-Fe-Ni signals.

***Magnesium-rich (Mg-rich)*** particles are considered to originate from sea salt (Dall'Osto & Harrison, 2006; L. Zhou et al.,
2006). The positive mass spectrum contains mainly cations such as Mg$^+$, Na$^+$, MgOH$^+$, Ca$^+$, CaOH$^+$, etc. The negative mass
spectrum contains, O$^-$, OH$^-$, Cl$^-$, CN, nitrate, sulfate, etc.



***Sodium-rich (Na-rich)*** particles are also thought to be derived from sea salt in many studies (Dall'Osto & Harrison, 2006; Healy et al., 2010; Köllner et al., 2017; L. Li et al., 2014; Shen et al., 2019). The positive signal may contain $Na^+$, $Mg^+$,

$NaO^+$, $Na_2^+$, $Na_2O^+$, $Na_2OH^+$ or $Na_2Cl^+$. The negative signal contains $O^-$, $OH^-$, $Cl^-$, nitrate, sulfate, $NaCl^-$ and $NaCl_2^+$, etc.

## 3.3 Mass spectra classification with supervised learning

To train the various models, we mapped the negative and positive mass spectra onto a 240-dimensional vector *X*. Before mapping, both mass spectra were normalized by their maximum peaks, respectively. Each vector element $X_i$ *(i = 1…240)* corresponds to a different mass-to-charge ratio (*m/z = -120…+120*), which is referred to as a "learning feature" and will be

fed into the model as an input variable. Fig. 2 illustrates the mapping of a mass spectrum to a vector and the workflow of classification.

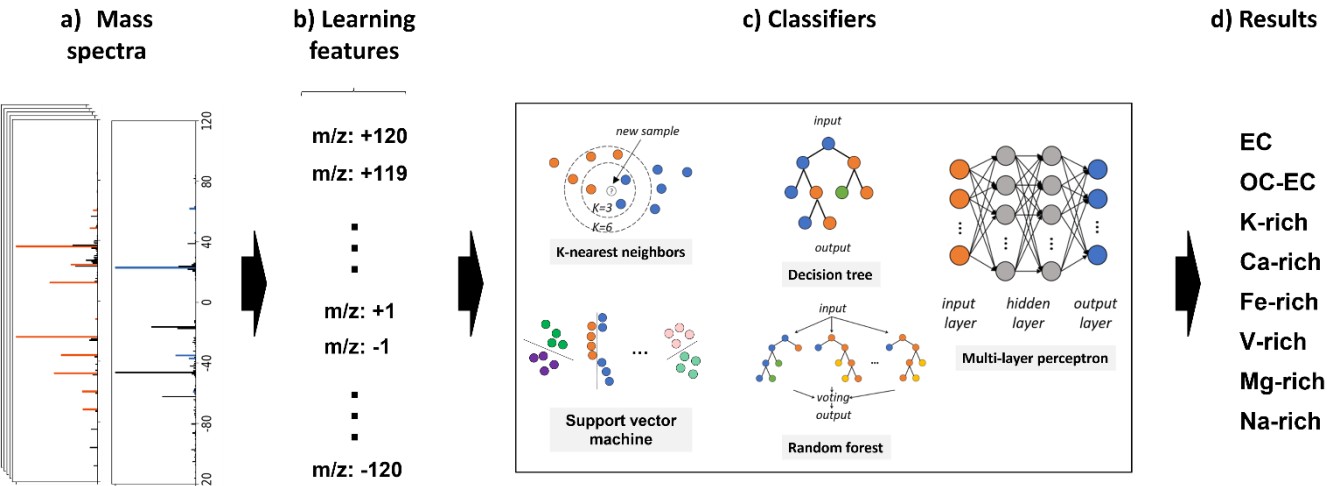

**Figure 2: Schematic diagram of the classification for SPMS data, from left to right: a) normalized mass spectra; b) each mass**
**spectrum is mapped onto a 240-dimensional vector, whose elements are the learning features for model training; c) and d)**
**different classification algorithms and resulting classes applied in this study.**

We randomly divided the 24,030 labeled mass spectra into three independent parts: training, validation and test set, in a ratio of 6:2:2 in terms of the number of samples, see Table 2. The training set is used to train the model parameters. The validation

set is used to check the state of the model during training, such as if the model is overfitting. In general, the training and validation sets are directly and indirectly involved in the training and tuning process and obviously do not reflect the actual capability of the model. Therefore, the model quality is evaluated using a test set. The training step is repeated until a model is obtained that performs satisfactorily in all three datasets. The experiments were performed with the following configurations: Windows 10, NVIDIA GeForce RTX 3090 graphics card, 3.2GHz Intel Core i9-12900K processor, and

64GB DDR3 RAM. We used Python 3.10 and the scikit-learn library to train K-NN, SVM, DT and RF models and a



machine learning framework PyTorch to train the neural network-based MLP model, taking advantage of GPU-accelerated computing. The Python-based libraries are open-source and cost-free. For tuning the parameters, we used a grid search strategy, which selects the best-performing combination of parameters as the final parameters of the model by looping over all predefined parameter values in the grid. Table 3 lists the optimal parameters for each algorithm. A detailed description

and discussion of the main parameters to be set by the user for the different algorithms can be found elsewhere (Awad & Khanna, 2015; Bishop, 2006; Mitchell, 1997).

**Table 3: List of optimal parameters for training different models as resulting from a grid-search strategy.**

| Method | Parameters | Training time (sec) |
|---|---|---|
| K-nearest neighbors | number of neighbors: 5<br>weight function: uniform<br>algorithm used to compute the nearest neighbors: auto | ~0 |
| Support vector machine | multi-class classification strategy:<br>one-vs-one<br>kernel: radial basis function | 0.6 |
| Decision tree | function to measure the quality of a split: gini<br>depth of the tree: none (unlimited) | 0.4 |
| Random forest | function to measure the quality of a split: gini<br>depth of the tree: none (unlimited)<br>number of trees in the forest: 110 | 2.0 |
| Multi-layer perceptron | number of hidden layers: 2<br>size of hidden layer: 512<br>learning rate: 0.001<br>number of iterations: 200<br>batch size: 1024<br>activation function: relu<br>loss function: cross entropy loss<br>solver for weight optimization: adam<br>dropout rate: 0.5 | 31.9 (GPU)<br><br>457.3 (CPU) |

## 4 Results

### 4.1 Metrics

Metrics are quantitative indicators to evaluate the performance of models. One evaluation metric can only reflect part of the model's performance, so different evaluation metrics might be selected for different application scenarios. The evaluation metrics used in this paper are *overall accuracy (OA)*, *precision*, *recall* and *F1-Score*.

***Overall accuracy (OA)***, also called prediction rate, is the most used evaluation metric, as it presents the ratio of correctly

classified samples to the total number of samples. However, in the case of imbalanced datasets, this metric has a serious drawback, since classes with large numbers of samples will affect *OA* the most.

***Recall and Precision*** are both fundamental metrics in case of imbalanced datasets. A trade-off between them usually requires optimizing one or the other depending on the application scenario. *Recall* states how many of all samples of *class A*



are actually predicted as belonging to it. *Precision* states what percentage of samples predicted to belong to *class A* actually

belongs to class A. The purpose of this work is to build a predictive model for automatic monitoring of individual particles, where the goal is to detect all aerosol particles of interest with both high *recall* and high *accuracy* are important.

*F1-Score* is the harmonic mean of *recall* and *precision*. Because they have a reciprocal relationship, optimizing one comes at the cost of the other. Therefore, a model to be both sensitive (high *recall*) and precise can be found using *F1-Score*.

*Confusion matrix* is a visualization tool, which compares the predicted labels with the actual labels of all given classes. In

our case of classifying mass spectra into one of eight classes, the *confusion matrix* is a *8 x 8* matrix with 64 elements, where the rows of the matrix refer to the actual labels and the columns refer to the predicted labels. The element in the *i-th* row and *j-th* column of the matrix indicate the number of samples actually labeled as class *i* and being predicted as class *j* (*i, j = 1...8*), with *1: EC, 2: OC-EC, 3: K-rich, 4: Ca-rich, 5: Fe-rich, 6: V-rich, 7: Mg-rich, and 8: Na-rich*. Therefore, the elements in the diagonal of the matrix (*i = j*) correspond to the number of correctly predicted samples; all remaining entries

(*i ≠ j*) are the numbers of incorrect predictions. For normalization, the matrix elements in each row are divided by the total number of predictions in that row and presented as percentage values, and finally, the normalized confusion matrix can be displayed by a heat map to visualize the percentage values, see Fig. 3.

## 4.2 Performance evaluation

For the five investigated supervised classification algorithms, the optimized models performed well, with *OA*, *recall*,

*precision* and *F1-Score* all above 94 % (some even above 97 %) for the 4,806 particle samples in the test set, see Table 4. The classification is performed fully automated, does not require any post-processing, and takes negligible time. In Fig. 3 the normalized confusion matrixes for each algorithm are displayed. It is observed that the K-NN and DT models have the lowest rates in all evaluation metrics and show significant misclassification for several classes. K-NN is less sensitive to subtle differences among the mass spectra. The DT model is prone to overfitting during training and has insufficient

generalization ability, while the performance of RF is significantly improved by using multiple DTs. RF, SVM and MLP all performed well and their evaluation metrics are similar. Studying the confusion matrices, some classes are obviously not as well identified as others. In the sequel, possible explanations are given, which may provide directions to improve our future work.

**Table 4: Overall accuracy, recall, precision and F1-Score comparison**

| Method | OA | Recall | Precision | F1-Score | Time |
|---|---|---|---|---|---|
| Test set: 4806 particles | | | | | |
| K-nearest neighbors | 95.3 | 94.5 | 94.1 | 94.3 | 0.2 |
| Support vector machine | **97.8** | **97.9** | 97.2 | 97.5 | 0.3 |
| Decision tree | 96.5 | 96.4 | 96.6 | 96.5 | **0.002** |
| Random forest | 97.6 | 97.4 | 97.5 | 97.5 | 0.1 |
| Multi-layer perceptron | 97.6 | 97.7 | **97.6** | **97.7** | 0.02 |







**Figure 3: Normalized confusion matrix. The numbers in the main diagonal correspond to the prediction accuracy of each class. All other entries indicate prediction errors e.g. the (2, 1) element in (a) indicates that 4.3 % of *OC-EC* particles were incorrectly predicted as *EC* particles.**


*EC* **vs** *OC-EC*. The (1, 2) and (2, 1) elements in the confusion matrix show that the particles from *EC* class and *OC-EC* class are occasionally misclassified. The mass spectra of EC particles are dominated by the EC signals, all other peaks being small. However, when other peaks become stronger (e.g. OC), EC will be no longer dominant in the mass spectrum, even though the highest peak may remain an EC peak, and such a spectrum should no longer be assigned to the *EC* class. But

delineating cases when the EC peaks will cease to be dominant is not easy to define. This might be illustrated by the two very similar mass spectra in Fig. 4 (a) and (b), whose highest and dominant peaks are clearly EC signals. The only difference is that the relative intensity of the $^{30}NO^+$ ion peak in the mass spectrum in Fig. 4 (b) is stronger than that in Fig. 4 (a). The mass spectrum (a) is labeled as *EC* class and the mass spectrum (b) is labeled as *OC-EC*. It is difficult to set the threshold of e.g. $^{30}NO^+$ that separates them, even for humans.





***OC-EC* vs *K-rich*.** The peak at *m/z = 39* can be either the signature signal K$^+$ of the *K-rich* class or the C$_3$H$_3^+$ ion which is common for the *OC* class. When the model encounters a *K-EC-OC* mass spectrum and an *OC-EC* mass spectrum, it sometimes predicts incorrectly (see Fig. 4 (c) and (d)).

***Fe-rich* vs *V-rich*.** Spectra of these two classes were sometimes predicted incorrectly due to difficulties to set a threshold value to separate them, similar to the *EC* and *OC-EC* classes. *V-rich* particles contain a combination of V-Fe-Ni ions, with

the Fe ion being sometimes the highest peak in the mass spectrum. Similarly, the highest ion peak of most mass spectra in the *Fe-rich* class is also from the Fe ion, and *Fe-rich* particles may also have very weak V$^+$ and Ni$^+$ signals. Two examples are shown in Fig. 4 (e) and (f).

**$^{39}$*K* vs $^{40}$*Ca* and $^{23}$*Na* vs $^{24}$*Mg*.** The marker peaks of the classes *K-rich* and *Ca-rich* as well as *Na-rich* and *Mg-rich* are separated by just one *m/z* (*39* vs. *40* and *23* vs. *24*, respectively). Therefore, these pairs of classes are prone to

misinterpretations by the algorithms. K-NN, which is distance-based for classification, has a significantly higher error rate in identifying such small differences between the spectra than the other four investigated algorithms. In some studies (Strehl et al., 2000; Zhong, 2005), researchers have noticed that Euclidean Distances are not well suited for the analysis of high-dimensional sparse data. Our experimental results validate this argument and clearly demonstrate that distance-based K-NN is less efficient than others in classifying high-dimensional SPMS data.

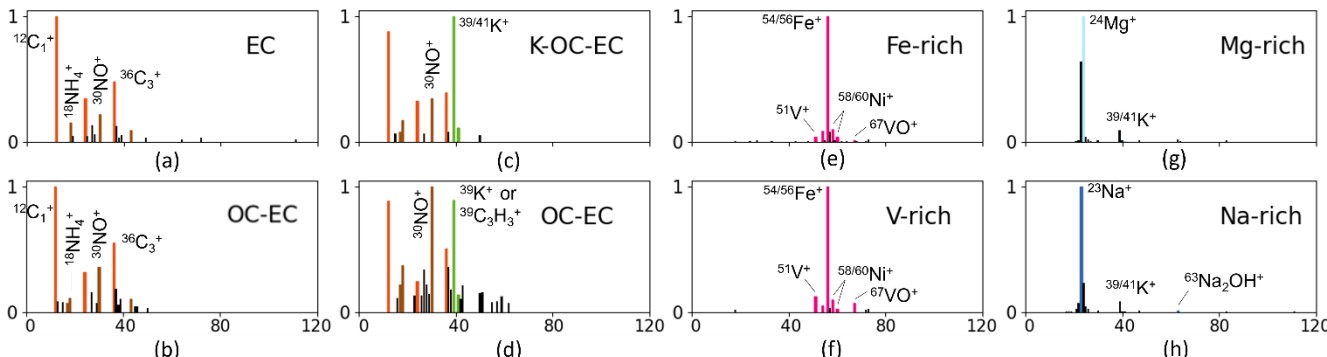


**Figure 4: The mass spectra in each column are easily confused with each other. The actual label is noted in the upper right corner on each mass spectrum.**

**4.3 Prediction of blind data**

The objective of this study is to provide a basis for real-time monitoring of air quality through the automatic classification of

SPMS data. Apparently, since there are much more types of aerosol particles in the air than the eight coarse classes described above, we expect the model to be able to distinguish particles that do not belong to the eight classes. One of the drawbacks of supervised learning algorithms, however, is that they generally cannot identify classes other than those present in the training data. As a solution, we use the predicted probability of the model to set a threshold value. Predicted probabilities below this threshold are assigned to an additional class "unclassified". By subsequently investigating the mass spectra in the



class "unclassified", we can later add newly discovered classes of interest to our dataset, to continuously expand the diversity of our dataset.

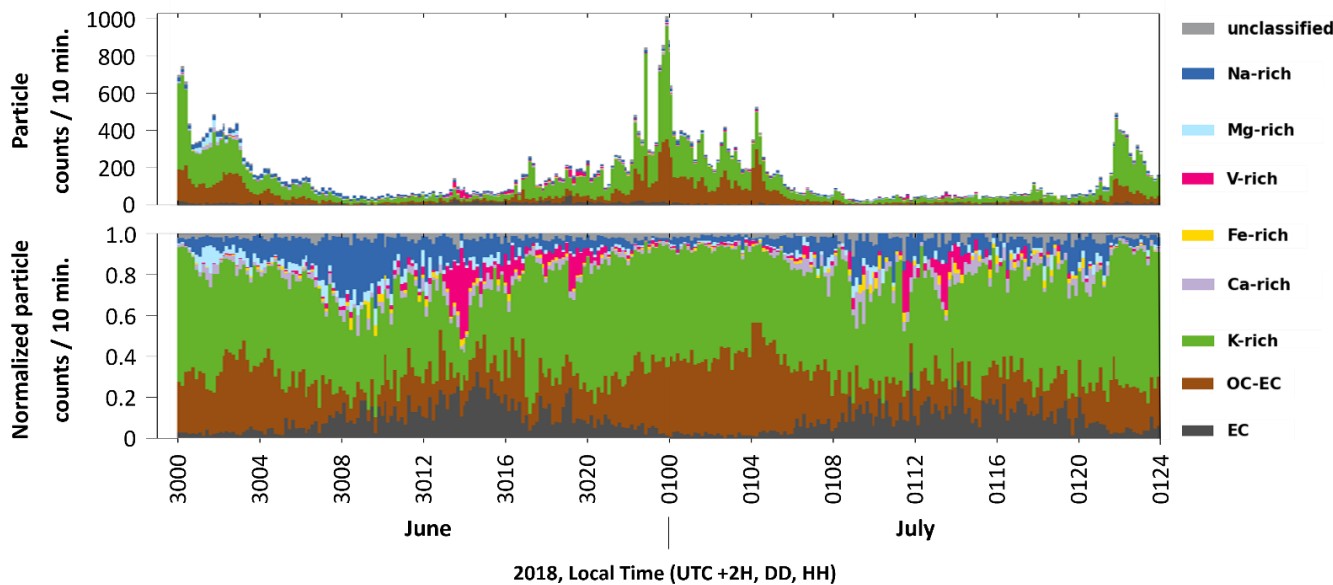

**Figure 5: Time distribution of aerosol particle classes predicted by the MLP model for data collected at the measurement site from 30 June to 1 July. The vertical axes of the plot above and below show the absolute and relative numbers of particles in every 10**
**minutes, respectively.**

As an example, the MLP algorithm uses the SoftMax function to compute the probability that the samples belong to different classes. Since the variables used in the SoftMax function are derived from the trained weights, the contingency caused by a winner-takes-all statistical approach (as for RF and SVM) can be considerably reduced. Therefore, the MLP model was
chosen to predict the class assignments of the "blind data" comprising two consecutive days (48 hours) of continuous measurements with 49,097 particles. Mass spectra with a predicted probability of less than 70 % were assigned to the "unclassified" class. Fig. 5 displays the resulting temporal distribution of the aerosol particle classes automatically predicted by MLP within only 0.2 seconds.

The class distribution over time shows a significant increase in the percentage of V-rich particles in the air around 14:00 and
19:00 on 30 June and around 12:00 and 14:00 on 1 July. A previous study (Passig et al., 2021), examining the same measurement data using the ART-2a algorithm, had found similar transients of V-rich particles in these time intervals and attributed them to ship passages. This consensus of our results proves the reliability of MLP supervised learning predictions.



## 5 Conclusions

In this study, new concepts for the automated classification of SPMS-analyzed aerosol particles using supervised learning
methods were described and the method's performance was evaluated on behalf of a dataset from a week-long summer
campaign in close vicinity to the Baltic Sea. Confronted with a lack of publicly available datasets of well-characterized
classified air-transported aerosol particles, we relied on mass spectra from published studies and expert knowledge to create
a dataset containing ~24,000 labeled mass spectra, each attributed to one of eight aerosol particle classes. As a result of this
time-consuming process, a well-characterized benchmark dataset of considerable size is now available for further SPMS
studies and liable to be publicly accessible, to open it up for further extensions. We used this dataset to train five models
popular for machine learning applications and compared their performance. All models performed well, with classification
accuracies of up to 97.8 %. In addition, we overcome the shortcoming that supervised learning cannot identify classes not
present in the training data. Based on a predicted probability of class assignment, mass spectra are classified to an additional
class of "unclassified" signature, liable to be later verified by an unsupervised, expert-supported algorithm. Finally, a neural
network-based MLP algorithm was used to automatically predict the "blind data" to acquire the temporal distribution of
aerosol particles, which makes it feasible to classify the measured data in real-time. Several advantages of using supervised
algorithms compared to unsupervised clustering algorithms could be proved. Besides the fact that - once a model is trained -
the classification becomes fully automated with processing times to classify tens of thousands of particles in less than a
second, the predictions are quantifiable through several evaluation metrics.

Supervised learning and unsupervised learning are two main categories of machine learning, and they differ based on the
type of input data and the problem they are solving. For the classification of aerosol spectra obtained by an SPMS
instrument, time for supervised learning is spent on dataset creation, as labeled datasets are expensive and time-consuming to
obtain. On the other hand, unsupervised clustering algorithms require time for post-processing, but this aspect could be
advantageous for discovering new particle classes, which is a limitation of all supervised learning algorithms. Overall,
supervised learning shows immense potential for real-time classification of SPMS data, particularly for the automatic
detection of specific particle classes. Our future work will involve expanding and diversifying our dataset, as well as
applying advanced and highly selective deep learning algorithms to enhance the generalization of the classification models.
This work, on the one hand, is a step towards rapid on-line classification of aerosols (mass or source fractions) and towards
future quantification routines using a parametrization with data from AMS (Aerodyne Aerosol Mass Spectrometry) and other
quantitative technologies. On the other hand, rapid classification of common particles reduces the remaining dataset and
enables target searches for hazard indicators (i.e. toxicants, airborne pathogens, explosives, drugs, industrial chemicals) and
thus supports the SPMS application in the field of hazard and air quality monitoring.



**Code availability**

The source code in Python as well as the trained models are available at:

https://github.com/GuanzhongLRT2/Machine-learning-approaches-for-automatic-classification-of-single-particle-mass-
spectrometry-data

**Data availability**

Data are available on request from Johannes Passig (johannes.passig@uni-rostock.de)

**Author contribution**

GW and HR designed the study, prepared the figures and wrote the manuscript with contributions from all the authors. GW
labeled the data, developed software and performed the computer experiments. JS and JP provided the SPMS data and
related knowledge. TA, GD and RZ assisted with technical support, data interpretation and manuscript writing.

**Competing interests**

The authors declare that they have no conflict of interest.

**Acknowledgements**

This research [project "LUKAS", https://dtecbw.de/home/forschung/unibw-m/projekt-lukas] is funded by dtec.bw –
Digitalization and Technology Research Center of the Bundeswehr. dtec.bw is funded by the European Union –
NextGenerationEU.

Funding by the Project "HazarDust" (German Federal Ministry for Education and Research, grant no. 13N15567, J.P., R.Z.)
is gratefully acknowledged.

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
