# Peer review of "Machine learning approaches for automatic classification of singleparticle mass spectrometry data"

_EGUsphere, 2023_

## Referee Comment (RC1)

General Comments:

The manuscript by Wang et al. focused on supervised algorithms to tackle the problem of automatic classification of large amounts of aerosol particle data. They created a labeled benchmark dataset containing ~24,000 particles with eight different coarse categories. They used the dataset from a week-long summer campaign to train five models popular for machine learning applications and compared their performance. The prediction accuracy of several supervised algorithms exceeded 97 % for the classification of the eight classes. The topic fits well within the scope of AMT. This manuscript is generally well written. Before its publication, the following comments need to be addressed.
.

Specific Comments:

Is there any specific reason for adopting June 29 as the demarcation between labeling and blind data? Please specify. There might be a significant uncertainty when choosing different demarcations. The authors need to address such uncertainties in the revised manuscript.

In Section 3, a brief description of the sampling lines, flow rate, and residence time for all measurements would be good. Further, it would be beneficial if the authors provide more details about the calibrations for the SPMS.

More evidence needs to be listed to support the the eight coarse particle classes used in this study. Why didn't the authors choose 7 or 9 coarse particle classes instead of 8? Some secondary particles are classified as primary emissions, e.g., K-rich particles and OC-EC particles. I also notice that V-rich particles contain 54/56Fe+ signal. Does it mean that the resolved V-rich particles were a mixed factor? How about combining the two factors together?

It would be better to add detailed comparisons (including time series and mass spectra) between the predictions and the the results obtained in a previous study using ART-2a. Otherwise, we don't know how significant the similarity variations are.

---

## Referee Comment (RC2)

Wang et al. studied approaches for the automatic classification of SPMS data by machine learning. The authors created a dataset containing 24,000 particles and used supervised algorithms to tackle the classification problem. Considering the importance of automatic classification of large amounts of SPMS data and the potential use of this automatic approach in future real-time monitoring, I would support the final publication after addressing the comments below.

Specific Comments

Section 2: The previous work should be highly summarized rather than listed one by one. I recommend that the summary of related work be concluded in the introduction. Some parts in Section 2, e.g., the description of different algorithms, should be elaborated in the methodology. The authors should better organize the structure of the manuscript.

Lines 71–73: Please rewrite the sentence. The numbers are presented misleadingly.

Line 175: Please add more details about the SPMS measurement and analysis, e.g., sensitivity, calibration, uncertainty, software, etc. SPMS also gives the particle size information. Could the authors provide more results about the particle size measurement? Will the particle size affect the automatic classification results?

Line 188: How do authors divide the data into two parts for labeling and verification? Are there any criteria, or are they just random? If the data are derived from the same sampling site, which means these particles probably have similar composition, is it reasonable to divide the data into different parts and use the "blind data" for verification?

Line 350: The authors mentioned the signal of $K^+$ and the signal of $C_3H_3^+$ at the same m/z position, which brought some uncertainty to the prediction. Is it possible to distinguish these ions at the same m/z position in SPMS? Could the authors estimate the uncertainties of applying the method used in this study to analyzing the SPMS data from other sites with different aerosol compositions?

Section 4.2: Since the optimized models with the five algorithms all performed well, which algorithm would the authors recommend in the future work? Now the prediction accuracy of supervised algorithms exceeded 97%. Will the accuracy still be perfect when the approaches are used for analyzing other datasets? How would the aerosol sources impact the prediction results? The authors should add more discussion on the uncertainty of the method, and the feasibility of application in other areas.

---

## Author Comment (AC1)

**Answers to Specific Comments:**

*Is there any specific reason for adopting June 29 as the demarcation between labeling and blind data? Please specify. There might be a significant uncertainty when choosing different demarcations. The authors need to address such uncertainties in the revised manuscript.*

MS data were recorded from June 26th to July 2nd, 2018, with only 2 hours of measurement on July 2nd, hence roughly 7 days in total. For simplicity, we arbitrarily chose the data from June 26th to June 30th to create the benchmark dataset and for training and the June 30-31 data for testing, as we did not note a severe time-dependence of the composition of the 8 chosen classes during the whole measurement campaign.

*In Section 3, a brief description of the sampling lines, flow rate, and residence time for all measurements would be good. Further, it would be beneficial if the authors provide more details about the calibrations for the SPMS.*

The SPMS instrument is a bipolar time-of-flight mass spectrometer (ATOF-MS) with an aerodynamic lens and an optical sizing unit. Detailed descriptions of its functionality can be found in (L. Li et al., 2011) and (Zhou et al., 2016). Briefly, for velocimetric particle sizing, two continuous wave lasers with a wavelength of 532 nm, ellipsoidal mirrors, and photomultipliers are employed. The compact mass spectrometer in Z-TOF geometry (Pratt and Prather, 2012), is equipped with continuous-wave 248.3 nm KrF excimer lasers used to detect the particles prior to LDI (L. Li et al., 2011; Passig et al., 2020; Schade et al., 2019; Y. Zhou et al., 2016). This wavelength is very well suited for resonance-enhanced laser desorption/ionization (LDI) of iron and other transition metals of interest for the analysis of ship exhaust particles in ambient air (Passig et al., 2020). The optical setup was optimized to achieve a hit rate of about 50% (#mass spectra/sized particles). The lens (f = 200 mm) is brought to an off-focus position of 7 mm relative to the particle beam, resulting in a spot size of 150 x 300 μm and an intensity of 5 GW cm$^{-2}$ at 6 mJ pulse energy (Passig et al., 2020; Schade et al., 2019). To analyse a sufficient number of particles, a multi-stage virtual impactor was used (Model 4240, MSP Corp., USA). From the 300 L min$^{-1}$ intake airflow, particles were concentrated into 1 L min$^{-1}$ carrier gas stream (6 x 4 mm conducting tube), from which 0.1 L min$^{-1}$ entered the SPMS instrument after a transfer time of few seconds. Monodisperse polystyrene particles were used for the size calibration of inlet and soot particles for the mass calibration of the mass spectrometer. Particle numbers were not corrected for size-dependent or type-dependent detection efficiencies (Shen et al., 2019).

*More evidence needs to be listed to support the eight coarse particle classes used in this study. Why didn't the authors choose 7 or 9 coarse particle classes instead of 8?*

Of note, the abundance of particles in different classes varies greatly, for example, particles in the K-rich and OC-EC classes make up about 80% of all particles measured. In order to obtain a benchmark dataset with comparable numbers of spectra in each class (ideally the same number of particles in each class, for a balanced dataset) we viewed a number of mass spectra much larger than the 24,000 mass spectra we eventually assigned to one of 8 classes ('labeled') for the results presented in this paper. The vast majority of these mass spectra belonged to the 8 classes. The number of particles not belonging to one of these 8 classes was very small and not enough to create separate classes for them in the dataset.

*Some secondary particles are classified as primary emissions, e.g., K-rich particles and OC-EC particles. I also notice that V-rich particles contain 54/56Fe+ signal.*

The main purpose of this paper is to verify whether supervised learning can accomplish the task of automated classification of SPMS data. Therefore, the 8 classes in this dataset were not divided further into subclasses (primary or secondary particles). Later, we will create refined datasets of labeled data, taken into account, for example, the degree of aging of particles emitted from ships, refining e.g. the Fe-rich class into Fe-EC, Fe-Sul, Fe-Nit, Fe-Nit-EC (primary and secondary particles).

***Does it mean that the resolved V-rich particles were a mixed factor? How about combining the two factors together?***

Indeed, many particles within the V-rich class showed a mixture with 54/56Fe. Specifically, the aerosol particles emitted from ships using heavy fuel oil (HFO) have shown a frequent combination of V-Fe-Ni ions.

***It would be better to add detailed comparisons (including time series and mass spectra) between the predictions and the results obtained in a previous study using ART-2a. Otherwise, we don't know how significant the similarity variations are.***

1) The results obtained with the ART-2a and ML-based approaches are not well comparable at the level of classification accuracy. With ART-2a, depending on the vigilance parameter (range from 0 to 1) a different number of clusters is generated. In most cases, for a practically feasible classification result, manual selecting and merging clusters are required. The smaller the vigilance factor, the fewer the number of generated clusters, and the lesser post-processing needed, but the accuracy will undoubtedly decrease. In the extreme case, when the vigilance is set to 1, each mass spectrum will form a different cluster and the classification accuracy would be 100 %, because we need to manually select and merge all of the clusters. Hence, with ART-2a heavily relying on manual post-processing, to compare its performance to that of automated classification algorithms would be not fair.

2) The main purpose of this paper is to verify whether supervised learning can be applied to the fully automated classification task of SPMS data, which cannot be achieved by ART-2a. A detailed comparison with previous studies using ART-2a (although, as motivated, difficult and case-dependent) would significantly increase the length of the text and make the topic of the paper less clear. We will consider taking up this comparison in a future paper specifically devoted to it, addressing also the balance between manual post-processing workload and the classification accuracy.

---

## Author Comment (AC3)

**Answers to Specific Comments:**

*Section 2: The previous work should be highly summarized rather than listed one by one. I recommend that the summary of related work be concluded in the introduction.*

Thank you for this comment and the recommendation. We are sorry to be not in full agreement with it. The intension of Section 2 was to give the reader a clearer understanding of the principles and various implementations of task-specific unsupervised learning used to classify sources of aerosols by providing a comprehensive review of the approaches and classification results published in previous studies.

We reviewed five different unsupervised learning algorithms (k-means, ART-2a and density-based clustering algorithms like DBSCAN and OPTICS), which were applied to SPMS data classification in the past. We belief that it would be interesting to briefly read about the main principles and differences between those approaches.

If these approaches were to summarize, we'd say that they are all characterized by the need of extra manual post-processing.

*Some parts in Section 2, e.g., the description of different algorithms, should be elaborated in the methodology. The authors should better organize the structure of the manuscript.*

We kindly disagree with that view. After having presented in Section 2 the foundations of five approaches of supervised learning algorithms generally applicable to SPMS data classification, in Section 3 we focus on our specific problem and describe the task-related methodology and important implementation steps of applying these supervised learning algorithms to our data.

*Lines 71–73: Please rewrite the sentence. The numbers are presented misleadingly.*

We agree. For clarification, the sentences were rewritten as follows:

In the field of SPMS data analysis, Arndt et al. (2021) and Healy et al. (2010, 2012) used the K-means algorithm in two rounds, to classify 558,740, 1.75 million and ~800,000 collected particles, respectively. First, these particles were pre-classified into 50, 80 and 80 different clusters, and subsequently, using the K-means algorithm again, these clusters were merged into 14, 15 and 33 different classes, respectively.

*Line 175: Please add more details about the SPMS measurement and analysis, e.g., sensitivity, calibration, uncertainty, software, etc.*

Since the first reviewer made the same suggestion, please allow us to use the same answer here.

The SPMS instrument is a bipolar time-of-flight mass spectrometer (ATOF-MS) with an aerodynamic lens and an optical sizing unit. Detailed descriptions of its functionality can be found in (L. Li et al., 2011) and (Zhou et al., 2016). Briefly, for velocimetric particle sizing, two continuous wave lasers with a wavelength of 532 nm, ellipsoidal mirrors, and photomultipliers are employed. The compact mass spectrometer in Z-TOF geometry (Pratt and Prather, 2012), is equipped with a 248.3 nm KrF excimer ionization laser. This wavelength is well suited for resonance-enhanced laser desorption/ionization (LDI) of iron and other transition metals (Passig et al., 2020) , e.g.

for the analysis of ship exhaust particles in ambient air (Passig et al., 2021). The optical setup was optimized to achieve a hit rate of about 50% (#mass spectra/sized particles). The lens (f = 200 mm) is brought to an off-focus position of 7 mm relative to the particle beam, resulting in a spot size of 150 x 300 µm and an intensity of 5 GW cm-2 at 6 mJ pulse energy (Passig et al., 2020; Schade et al., 2019). From the 300 L min$^{-1}$ intake airflow, particles were concentrated into 1 L min$^{-1}$ carrier gas stream (6 x 4 mm conducting tube), from which 0.1 L min$^{-1}$ entered the SPMS instrument after a transfer time of few seconds. Monodisperse polystyrene particles were used for the size calibration of inlet and soot particles for the mass calibration of the mass spectrometer. No corrections were made for size-dependent or type-dependent detection efficiencies (Shen et al., 2019).

(will be added in Lines 173-177)

*SPMS also gives the particle size information. Could the authors provide more results about the particle size measurement? Will the particle size affect the automatic classification results?*

Due to the wavelength (532 nm) of the two continuous wave lasers of the sizing unit, the lower boundary of measurable particle sizes is approx. 150-200 nm. With the SPMS instrument, particle of sizes up to 2,5 µm can be measured.

(will be added in Lines 173-177 of the manuscript)

In this work, we used only the chemical composition of the particles as learning features to train the classification model. Therefore, the sizes of the particles have no effect on the classification results.

(will be added in Line 275 of the manuscript)

*Line 188: How do authors divide the data into two parts for labeling and verification? Are there any criteria, or are they just random?*

Since the first reviewer asked the same question, please allow us to use a similar answer here.

SPMS data were recorded from 26 June to 02 July, 2018, with only 2 hours of measurement on 02 July, hence roughly 7 days in total. For simplicity, we arbitrarily chose the data from 26 June to 30 June to create the benchmark dataset and for training and the 30-31 June data for testing (blind data), as we did not note a severe time-dependence of the composition of the 8 chosen classes during the whole measurement campaign.

(will be added in Line 189 of the manuscript)

*If the data are derived from the same sampling site, which means these particles probably have similar composition, is it reasonable to divide the data into different parts and use the "blind data" for verification?*

We regard this as reasonable and proper practice. If measurement from a different sampling site were chosen are chosen as blind test data, this could be problematic, since to recognize data from different sources is still a common problem in machine learning. The professional term of this problem is robustness. That is, for verification, the data used for training and (blind) testing should have the same source (environment, measurement sampling site). One

way to improve robustness is to expand the labeled dataset with data from different sources to be used to train the model.

**Line 350: The authors mentioned the signal of K+ and the signal of C3H3+ at the same m/z position, which brought some uncertainty to the prediction. Is it possible to distinguish these ions at the same m/z position in SPMS?**

With pre-processed, quantized and normalized mass spectra, as we use them for labeling, ions at the same m/z position would not be directly distinguishable. A distinction could be based on the combination of the chemical composition ("the peak pattern" as the intensity of all other peaks in the mass spectrum), the particle size and the abundance (probability of occurrence).

**Could the authors estimate the uncertainties of applying the method used in this study to analyzing the SPMS data from other sites with different aerosol compositions?**

This is subject of our current investigations. First results have been published recently, see (Wang et al., IEEE Sensors Letters 2023), made public in https://ieeexplore.ieee.org/document/10251644 (Early Access): In this study we created a new labeled dataset (37,406 particles within 13 classes). An overall classification accuracy of over 90% was achieved using a neural network based algorithm. (The accuracy is lower than that in this paper, but more particle classes were recognized).

**Section 4.2: Since the optimized models with the five algorithms all performed well, which algorithm would the authors recommend in the future work?**

We have already discussed the reasons for choosing MLPs in the text (please look at the following copy from the text). Furthermore, in our current research we found that neural network based algorithms like MLP perform better on a wider range of datasets from different seasons, sampling sites etc., i.e. the NN based algorithms are more robust. (This will be added to the manuscript.)

Advantages of MLP and major drawbacks of the other investigated algorithms were already summarized in the text and are repeated here.

From Lines 349-352

It is observed that the K-NN and DT models have the lowest rates in all evaluation metrics and show significant misclassification for several classes. K-NN is less sensitive to subtle differences among the mass spectra. The DT model is prone to overfitting during training and has insufficient generalization ability, while the performance of RF is significantly improved by using multiple DTs.

From Lines 358-364

$^{39}$K vs $^{40}$Ca and $^{23}$Na vs $^{24}$Mg. The marker peaks of the classes K-rich and Ca-rich as well as Na-rich and Mg-rich are separated by just one m/z (39 vs. 40 and 23 vs. 24, respectively). Therefore, these pairs of classes are prone to misinterpretations by the algorithms. K-NN, which is distance-based for classification, has a significantly higher

error rate in identifying such small differences between the spectra than the other four investigated algorithms. In some studies (Strehl et al., 2000; Zhong, 2005), researchers have noticed that Euclidean Distances are not well suited for the analysis of high-dimensional sparse data. Our experimental results validate this argument and clearly demonstrate that distance-based K-NN is less efficient than others in classifying high-dimensional SPMS data.

From Lines 382-386

As an example, the MLP algorithm uses the SoftMax function to compute the probability that the samples belong to different classes. Since the variables used in the SoftMax function are derived from the trained weights, the contingency caused by a winner-takes-all statistical approach (as for RF and SVM) can be considerably reduced. Therefore, the MLP model was chosen to predict the class assignments of the "blind data" comprising two consecutive days (48 hours) of continuous measurements with 49,097 particles.

*==Now the prediction accuracy of supervised algorithms exceeded 97%. Will the accuracy still be perfect when the approaches are used for analyzing other datasets?==*

This question again refers to the robustness of supervised ML algorithms. If we want the trained model to perform well across different datasets, we have to keep expanding the set of labeled data.
(This notion will be added to the manuscript.)

*==How would the aerosol sources impact the prediction results?==*

The particle classes predicted by supervised learning depend on the classes contained in the training dataset, i.e., classes other than in the training dataset cannot be recognized, see line 157.

*==The authors should add more discussion on the uncertainty of the method, and the feasibility of application in other areas.==*

Differences in measurement campaigns (SPMS instruments, sampling locations, weather conditions, etc.) lead to differences in data (mass spectra) from one measurement to another, even for particles of the same class. Models trained with data from a single measurement do not generalize well and are more sensitive to differences in the characteristics of data from different sources. Therefore, we need to expand our labeled dataset in future work. Through that, the generalization of the trained models will become stronger and stronger. In one of our current studies (*In: Proceedings of the Joint TAP & SE conference, Gothenburg, to be published in Oct. 2023*) we show that the robustness of a model trained using two merged independent datasets from different measurement campaigns is substantially improved and outperforms a model trained separately using one of the two datasets.
(will be added to the manuscript.)